# DEEP Q-NETWORK WITH PROXIMAL ITERATION

## ABSTRACT

We employ Proximal Iteration for value-function optimization in reinforcement learning. Proximal Iteration is a computationally efficient technique that enables us to bias the optimization procedure towards more desirable solutions. As a concrete application of Proximal Iteration in deep reinforcement learning, we endow the objective function of the Deep Q-Network (DQN) agent with a proximal term to ensure that the online-network component of DQN remains in the vicinity of the target network. The resultant agent, which we call DQN with Proximal Iteration, or DQNPro, exhibits significant improvements over the original DQN on the Atari benchmark. Our results accentuate the power of employing sound optimization techniques for deep reinforcement learning.

## 1 INTRODUCTION

A core competency of a reinforcement learning agent is the ability to learn in environments with large state spaces, which is key to solving important applications such as robotics (Kober et al., 2013), dialog systems (Williams et al., 2017), and large-scale games (Tesauro, 1994; Silver et al., 2017). Recent breakthroughs in deep reinforcement learning have demonstrated that existing algorithms such as Q-learning (Watkins and Dayan, 1992; Sutton and Barto, 2018) can surpass human-level performance on large-scale problems when they are equipped with deep neural networks for function approximation (Mnih et al., 2015).

Two key components of a gradient-based deep reinforcement learning agent are its objective function, and an optimization procedure. The optimization procedure takes estimates of the gradient of the objective with respect to network parameters, and updates the network parameters accordingly. In DQN (Mnih et al., 2015), for example, the objective function is the empirical expectation of the one-step temporal difference (TD) error (Sutton, 1988) on a buffered set of environmental interactions (Lin, 1992), and advanced variants of stochastic gradient descent are employed to update the network to best minimize this objective function.

A fundamental difficulty in minimizing this objective function stems from operating in what is known as the *deadly triad* (Tsitsiklis and van Roy, 1997; Sutton and Barto, 2018; van Hasselt et al., 2018): Updates are performed (1) off-policy and in conjunction with (2) non-linear function approximation and (3) bootstrapping. In the context of reinforcement learning, bootstrapping refers to the dependence of the target of our gradient updates on the parameters of the neural network, which is itself continuously updated during training. Employing bootstrapping in reinforcement learning stands in contrast to typical supervised-learning techniques, or even Monte-Carlo reinforcement learning (Sutton and Barto, 2018), where the target of our gradient updates does not directly depend on the parameters of the neural network.

Mnih et al. (2015) proposed a simple and intriguing approach to hedging against issues that arise when operating in the deadly triad, specifically to use a *target network* for bootstrapping in value-function optimization. In this case, the target network is updated periodically, and tracks the online network with some delay. While this modification constituted a major step towards combating mis-behavior in Q-learning (Lee and He, 2019; Kim et al., 2019; Zhang et al., 2021), extreme volatility in the optimization procedure is still prevalent. To ease optimization, we can synchronize the target and the online network less frequently, but doing so could be thought of as delaying policy improvement, therefore making the ultimate performance sensitive to the period of synchronization selected (Kim et al., 2019).

Our contribution is to employ Proximal Iteration to improve the stability of the optimization procedure in DQN. In particular, by observing a synergy between DQN and Proximal Iteration, we endow the objective function in DQN with a proximal term that ensures the parameters of the online-network component of DQN remain in the vicinity of the parameters of the target network. This novel combination adds no additional computational cost to the original DQN agent. We present comprehensive experiments on Atari benchmarks (Bellemare et al., 2013) demonstrating that our new algorithm, which we refer to as Deep Q-Network with Proximal Iteration or DQNPro, yields significant improvements over its vanilla DQN counterpart, thus revealing the benefits of combining deep RL algorithms with Proximal Iteration.

## 2 BACKGROUND ON REINFORCEMENT LEARNING

Reinforcement learning (RL) is the study of the interaction between an environment and an artificial agent that learns to maximize reward through experience. The Markov decision process (Puterman, 1994), or MDP, is used to mathematically define the RL problem. An MDP is specified by the tuple $\langle \mathcal{S}, \mathcal{A}, \mathcal{R}, \mathcal{P}, \gamma \rangle$, where $\mathcal{S}$ is the set of states and $\mathcal{A}$ is the set of actions. The functions $\mathcal{R} : \mathcal{S} \times \mathcal{A} \to \mathbb{R}$ and $\mathcal{P} : \mathcal{S} \times \mathcal{A} \times \mathcal{S} \to [0, 1]$ denote the reward and transition dynamics of the MDP. Finally, by discounting future rewards, $\gamma$ formalizes the intuitive notion that short-term rewards are more valuable than those received later.

The goal in the RL problem is to learn a policy, a mapping from states to a probability distribution over actions, $\pi : \mathcal{S} \to \mathcal{P}(\mathcal{A})$, that obtains high sums of future discounted rewards. An important concept in RL is the state–action value function ($Q$-function). Formally, it denotes the expected discounted sum of future rewards when taking a particular action $a$ in state $s$ and following policy $\pi$ thereafter:

$$Q^\pi(s, a) := \mathbb{E}\big[R_{t+1} + \gamma R_{t+2} + \gamma^2 R_{t+3} + \ldots \big| S_t = s, A_t = a, \pi \big] .$$

Define $Q^\star$ as the optimal value of a state–action pair: $Q^\star(s, a) := \max_\pi Q^\pi(s, a)$. The Bellman optimality operator $\mathcal{T} : \mathbb{R}^{|\mathcal{S}| \times |\mathcal{A}|} \to \mathbb{R}^{|\mathcal{S}| \times |\mathcal{A}|}$ is defined as follows:

$$\big[\mathcal{T}(Q)\big](s, a) := \mathcal{R}(s, a) + \sum_{s' \in \mathcal{S}} \gamma \, \mathcal{P}(s, a, s') \max_{a'} Q(s', a') .$$

$Q^\star(s, a)$ is the unique fixed-point of $\mathcal{T}$, meaning $Q^\star := \mathcal{T}(Q^\star)$. The Bellman operator is at the heart of many planning and RL algorithms such as Value Iteration (Bellman, 1957), which proceeds as follows given an arbitrarily initialized $Q_0$:

$$Q_{k+1} \leftarrow \mathcal{T}(Q_k) .$$

This iterative process is guaranteed to converge to the unique fixed point of $\mathcal{T}$, namely $Q^\star$, because $\mathcal{T}$ is a $\gamma$-contraction in the infinity norm:

$$\|\mathcal{T}(Q) - \mathcal{T}(Q')\|_\infty \leq \gamma \, \|Q - Q'\|_\infty .$$

In large-scale settings, it is not tractable to learn a separate number for each state–action pair. A common way to address this curse of dimensionality is to learn a representation of the Q function in the form of a parameterized function approximator, such as a deep neural network: $Q(s, a; w) \approx Q^\star(s, a)$, where $w$ here denotes the parameters of the function approximator. Notice that the Bellman operator may no longer be a contraction in the presence of general function approximation.

## 3 THEORY

Define the Bregman divergence generated by the convex function F as follows:

$$D_F(x, y) = F(x) - F(y) - \nabla F(y)^\top (x - y) ,$$

where we assume that $F$ is smooth in the sense that the gradient of $F$ is non-expansive:

$$||\nabla F(x) - \nabla F(y)|| \leq ||x - y|| .$$

We define the Bergman-smoothed Bellman operator, $T_D$ as follows:

$$T_D(v) = \arg\min_{v'} \frac{1}{2}||T(v) - v'||^2 + \frac{1}{2c}D_F(v', v) .$$

Define $\bar{v} = T_D(v)$ , we have:

$$\bar{v} = T(v) + \frac{1}{2c}\big(\nabla F(v) - \nabla F(\bar{v})\big),$$

We can then show:

$$
\begin{aligned}
||T_D(v_1) - T_D(v_2)|| &= ||\bar{v}_1 - \bar{v}_2|| \\
&= ||T(v_1) + \frac{1}{2c}\big(\nabla F(v_1) - \nabla F(\bar{v}_1)\big) - T(v_2) - \frac{1}{2c}\big(\nabla F(v_2) - \nabla F(\bar{v}_2)\big)|| \\
&\leq ||T(v_1) - T(v_2)|| + \frac{1}{2c}||\nabla F(v_1) - \nabla F(v_2)|| + \frac{1}{2c}||\nabla F(\bar{v}_1) - \nabla F(\bar{v}_2)|| \\
&\leq ||T(v_1) - T(v_2)|| + \frac{1}{2c}||\nabla F(v_1) - \nabla F(v_2)|| + \frac{1}{2c}||\bar{v}_1 - \bar{v}_2||
\end{aligned}
$$

This implies:

$$
\begin{aligned}
\frac{2c-1}{2c}||T_D(v_1) - T_D(v_2)|| &\leq ||T(v_1) - T(v_2)|| + \frac{1}{2c}||\nabla F(v_1) - \nabla F(v_2)|| \\
&\leq ||T(v_1) - T(v_2)|| + \frac{1}{2c}||v_1 - v_2|| \\
&\leq \frac{2c\gamma}{2c-1}\gamma||v_1 - v_2|| + \frac{1}{2c-1}||v_1 - v_2|| \\
&= \frac{2c\gamma + 1}{2c-1}||v_1 - v_2||,
\end{aligned}
$$

allowing us to conclude that:

$$||T_D(v_1) - T_D(v_2)|| \leq \frac{\gamma + 2c}{2c-1}||v_1 - v_2||$$

Therefore, the operator $T_D$ is a $\frac{\gamma+2c}{2c-1}$-contraction so long as $c > \frac{1}{1-\gamma}$ .

## 4 PROXIMAL OPERATOR

Consider the following optimization problem:

$$\text{minimize}_w \; h(x; w) , \tag{1}$$

where $h$ is the objective function (e.g., standard temporal difference (TD) error in reinforcement learning (5)), $w$ denotes the learnable parameters (e.g., the parameters of a Q-network), and $x$ is the input data (e.g., $\langle s, a, r, s'\rangle$). If $h$ is differentiable, then stochastic gradient descent (SGD), which proceeds as follows:

$$w_k \leftarrow w_{k-1} - \alpha_k \nabla_w h(x; w_{k-1}), \quad k = 1, 2, 3, \dots \tag{2}$$

could be used to solve the optimization problem (1). Here, $\alpha_k$ denotes the step size at iteration $k$, and $w_0$ is an arbitrary starting point[1].

While SGD is well-behaved asymptotically and under standard assumptions, it can perform recklessly in the interim, leading into a situation where the iterates get exponentially far from the solution (Moulines and Bach, 2011; Ryu and Boyd, 2014). This behavior can be particularly disruptive in the context of RL where sample efficiency is an important concern.

One simple and effective approach to stabilizing SGD is to augment the objective with a quadratic regularizer to ensure that the solution at each iteration stays in the vicinity of the previous ones:

$$w_k = \arg\min_w h(w) + \frac{1}{2c}\|w - w_{k-1}\|_2^2 . \tag{3}$$

---

[1]To simplify the notation and whenever it is clear from the context, we drop $x$ from the notation, i.e., $h(w) \triangleq h(x; w)$ and $\nabla h(w_{k-1}) \triangleq \nabla h(x; w_{k-1})$.

The quadratic term is basically a proximal term that ensures the next iterate $w_k$ will not stray too far from the previous iterate $w_{k-1}$. Having completed iteration $k$, we replace the previous iterate $w_{k-1}$ by its new value $w_k$ and then proceed to iteration $k+1$.

To understand the behavior of this iterative process, called Proximal Iteration (Parikh and Boyd, 2014; Ryu and Boyd, 2014), we define the proximal operator:

$$\text{prox}_h(w_k) := \arg\min_w \ h(w) + \frac{1}{2c}\|w - w_{k-1}\|_2^2 \ , \tag{4}$$

Proximal Iteration could be written as:

$$w_k \leftarrow \text{prox}_h(w_{k-1}) \quad k = 1, 2, 3 \ldots$$

From the Banach fixed-point theorem, we know that iteratively applying $\text{prox}_h$ eventually converges to a unique fixed point provided that the operator is a contraction. Bauschke et al. (2012) have shown that, if the function $h$ is $\mu$-strongly convex, then the operator $\text{prox}_h$ is indeed a contraction:

$$\|\text{prox}_h(w) - \text{prox}_h(w')\| \leq \frac{1}{1+\mu} \|w - w'\| \ ,$$

and therefore iteratively applying the operator would guarantee convergence.

What is the fixed-point of this operator? It is possible to show that it is the minimum of the function $h$, i.e., $\arg\min_w h(w)$, the minimum of problem (1), which we originally sought to solve (Bauschke and Combettes, 2011).

Therefore, the intuitive way of thinking about the Proximal Iteration algorithm, which is sometimes referred to as disappearing Tikhonov regularization (Parikh and Boyd, 2014), is as a principled way to bias the optimization path towards the previous iterate using the quadratic-norm penalty without altering the final answer. In other words, as Proximal Iteration makes progress, the previous iterate $w_{k-1}$ gets closer to $w^\star = \arg\min h(w)$ in light of the contraction property, therefore the contribution of the quadratic term vanishes to zero thus not biasing the asymptotic solution. The following remark sheds more light on the relationship between Proximal Iteration and trust-region algorithms:

**Remark 1** (**Relationship between trust region and Proximal Iteration**). *Define the trust-region problem as follows:*

$$\text{minimize}_w \ h(w)$$
$$s.t. \ \|w - w_{k-1}\|_2 \leq \rho \ ,$$

*Parikh and Boyd (2014) show that solving the above trust-region formulation is equivalent to a step of Proximal Iteration under appropriate choices of $\rho$ (above) and $c$ (in prox). Therefore, iteratively applying the two formulations leads to the same solution. (See their Section 3.4 for more details.)*

The second remark highlights the strength of Proximal Iteration relative to SGD:

**Remark 2** (**The non-asymptotic advantage of Proximal Iteration over SGD**). *While the asymptotic performance of SGD and Proximal Iteration are analogous (Ryu and Boyd, 2014), they may exhibit qualitatively different behavior in a non-asymptotic sense. More concretely, denote the step-size of SGD and Proximal Iteration at iteration $k$ by $\alpha_k = \frac{C}{k}$, and $w^\star := \arg\min_w h(w)$. For SGD, we have (Bauschke and Combettes, 2011):*

$$\text{E}\big[\|w_k - w^\star\|_2^2\big] \leq \mathcal{O}\left(\frac{exp(C^2)}{k^C} + \frac{1}{k}\right) \ .$$

*The exponential dependence on $C$ can indeed manifest itself as Ryu and Boyd (2014) demonstrate. In contrast, for Proximal Iteration we have (Ryu and Boyd, 2014):*

$$\text{E}\big[\|w_k - w^\star\|_2^2\big] \leq \|w_0 - w^\star\|_2^2 + \sum_{i=1}^{k} \alpha_i \ .$$

*While the iterates of SGD can get exponentially far from the optimal answer (Moulines and Bach, 2011; Ryu and Boyd, 2014), Proximal Iteration is fairly well-behaved in the interim. This property is particularly valuable in the context of reinforcement learning where sample-efficiency, and not just the asymptotic performance, is of utmost importance.*

In the context of a parameterized $Q$-function in RL, commonly used loss functions are not convex; however, several works have introduced convex loss via linear approximation. Lee and He (2019) have shown that in linear policy evaluation the loss function is convex in the weights. Serrano et al. (2021) introduced a convex loss function for control with linear approximation. These works demonstrate that many useful convex minimization objectives exist for RL.

In the case of deep RL, the loss function is not convex, and few convergence results exist for most state-of-the-art algorithms. Nevertheless, we follow standard practice and show that Proximal Iteration achieves significant empirical improvements when applied to standard deep RL algorithms. Furthermore, recent work provides convergence results for Proximal Iteration in specific non-convex cases, such as when $h(w)$ can be decomposed into a sum of a smooth non-convex function and non-smooth convex function (Fukushima and Mine, 1981; J Reddi et al., 2016; Li and Li, 2018). These recent results open the door for potential future developments toward proving convergence for Proximal Iterations in RL, which would further strengthen its theoretical grounding.

## 5   DEEP Q-NETWORK WITH PROXIMAL ITERATION

In this section, we introduce a new deep RL algorithm by applying Proximal Iteration to DQN. Recall that the original DQN agent employs two parameterized value functions: an online network $Q(s, a; w)$ that is updated at each step, and a target network $Q(s, a; \theta)$ that is used for bootstrapping and is synchronized with the online network periodically. More formally, let tuples $\langle s, a, r, s' \rangle$ denote the buffered environmental interactions of the RL agent. Define the following objective function akin to DQN:

$$h(w) := (1/2)\widehat{\mathbb{E}}_{\langle s,a,r,s' \rangle}\left[\left(r + \gamma \max_{a'} Q(s', a'; \theta) - Q(s, a; w)\right)^2\right]. \tag{5}$$

Given a fixed target-network weight $\theta$, in the original DQN, our desire is to find a setting of weights $w$ that minimizes $h$. We can do so by applying SGD, but as argued above, it can lead to reckless updates of $w$. To hedge against this possibility, we can bias the online weights towards the previous iterate. The target network is a natural choice for this purpose, motivating the following objective:

$$\arg \min_w h(w) + \frac{1}{2c} \|w - \theta\|_2^2 = \text{prox}_h(\theta) .$$

To solve this minimization problem, we can take multiple descent steps using the stochastic gradients of the objective. Thus, starting from the initial point $w = \theta$, we perform multiple $w$ updates (specified by the *period* hyper-parameter) as follows:

$$
\begin{aligned}
&w = \theta \\
&\text{for i} = 1 \dots \textit{period}: \\
&\quad w \leftarrow w - \alpha\big(\nabla h(w) + \frac{1}{c}(w - \theta)\big) \\
&\quad \theta \leftarrow w.
\end{aligned}
\tag{6}
$$

In the last step, akin to the original DQN, we synchronize the two networks by performing $\theta \leftarrow w$, before moving to the following iteration. This iterative optimization proceeds until convergence is obtained.

Observe that the online-network update (6) can equivalently be written as:

$$w \leftarrow \big(1 - (\alpha/c)\big) \cdot w + (\alpha/c) \cdot \theta - \alpha \nabla h(w) . \tag{7}$$

Notice the intuitively appealing form of the update: We first compute a convex combination of $\theta$ and $w$, based on the hyper-parameters $\alpha$ and $c$, then add the gradient term to arrive at the next iterate of $w$. If $w$ and $\theta$ are close, the convex combination is close to $w$ itself and so this DQNPro update would in effect perform an update similar to that of the original DQN. However, when the online weight $w$ strays too far from the previous target-network iterate, taking the convex combination ensures that the online network gravitates towards the target network by default. In other words, the gradient signal from minimizing the squared TD error (5) needs to be strong enough to compensate

---

**Algorithm 1** Deep Q-Network with Proximal Iteration (DQNPro)

---

1: Initialize $\theta$, N, *period*, replay buffer $\mathcal{D}, \alpha$, and $c$
2: $s \leftarrow$ env.reset(), $w \leftarrow \theta$, numUpdates $\leftarrow 0$
3: **repeat**
4:     $a \sim \epsilon\text{-greedy}\big(Q(s, \cdot; w)\big)$
5:     $s', r \leftarrow$ env.step$(s, a)$
6:     add $\langle s, a, r, s' \rangle$ to $\mathcal{D}$
7:     **if** $s'$ is terminal **then**
8:         $s \leftarrow$ env.reset()
9:     **end if**
10:    **for** $n$ in $\{1, \ldots, \text{N}\}$ **do**                  $\triangleright$ perform $N$ online-network updates
11:        sample a mini-batch $\mathcal{B} = \{\langle s, a, r, s' \rangle\}$ from $\mathcal{D}$

$$\nabla h(w) \leftarrow \nabla_w \frac{1}{2|\mathcal{B}|} \sum_{\langle s,a,r,s'\rangle \in \mathcal{B}} \big(r + \gamma \max_{a'} Q(s', a'; \theta) - Q(s, a; w)\big)^2$$

$$w \leftarrow \big(1 - (\alpha/c)\big) \cdot w + (\alpha/c) \cdot \theta - \alpha \nabla_w h(w)$$

$$\text{numUpdates} \leftarrow \text{numUpdates} + 1$$

12:        **if** numUpdates $\%$ *period* $= 0$ **then**           $\triangleright$ update the target network
13:           $\theta \leftarrow w$
14:        **end if**
15:     **end for**
16: **until** convergence

---

for the default gravitation towards $\theta$. Notice also that the update rule includes the original DQN update as a special case when $c \to \infty$.

The pseudo-code for DQNPro is presented in Algorithm 1. The difference between DQN (Mnih et al., 2015) and our DQNPro is minimal, and is highlighted in **red**. Note that our proposed method adds almost no additional computational cost to the original DQN. While the particular form of DQNPro presented in Algorithm 1 uses SGD as an optimizer, we note that the general idea behind our method can be applied with other forms of first-order gradient-based optimizers such as ADAM (Kingma and Ba, 2015), ADAGRAD (Duchi et al., 2011), etc.

## 6   EXPERIMENTS

In this section, we first evaluate the DQNPro agent relative to the vanilla DQN counterpart on a set of Atari-2600 benchmarks (Bellemare et al., 2013), and show that endowing the agent with the proximal term can lead to significant improvements in overall performance. We next investigate the utility of our proposed proximal term through ablation analyses and experiments. See also Appendix A for a complete description of our experimental pipeline, as well as Appendix B, and Appendix C for more results and analyses.

### 6.1   SETUP

We used 40 games from the Atari-2600 benchmark suite (Bellemare et al., 2013) to conduct our experimental evaluations. Following Machado et al. (2018) and Castro et al. (2018), our experiments used sticky actions to inject stochasticity into the otherwise deterministic Atari-2600 emulator.[2]

In our experiments, we endow DQN (Mnih et al., 2015) and DDQN (van Hasselt et al., 2016) with the proximal term and compare with the vanilla counterparts. Our training and evaluation protocols and the hyper-parameter settings closely follow those of the Dopamine[3] (Castro et al., 2018) baseline, which is consistent with the existing literature. To report performance results, we measure the

---

[2] When sticky actions are enabled, the emulator will ignore the agent's current action choice and execute the agent's previous action with small probability (Machado et al., 2018).

[3] https://github.com/google/dopamine

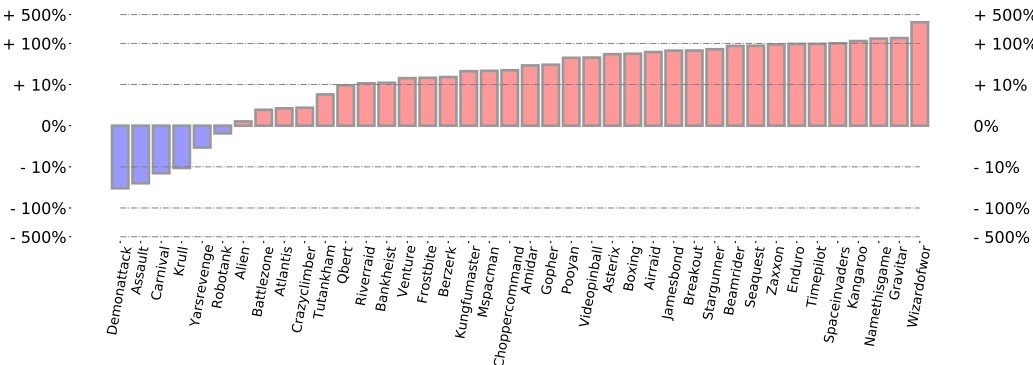

Figure 1: **Final performance gain for DQNPro over vanilla DQN.** Results are averaged over four random seeds, and the y-axis is scaled logarithmically.

undiscounted sum of rewards obtained by the learned policy during evaluation. We further report the learning curve for all experiments averaged across 4 random seeds in Appendix C. We reiterate that we used the same network architecture as that of Mnih et al. (2015) for all experiments in this paper and we also used the exact same hyper-parameters for all agents to ensure a sound and fair comparison. More details are provided in Appendix A.

We emphasize that our DQNPro only adds a single hyper-parameter, namely $c$, to the original DQN. We set the hyper-parameter $c = 0.2$ throughout all experiments with DQNPro and in all games. We did a minimal random search on four games, namely Asterix, Breakout, Qbert, and Seaquest, to tune this value. We also observed no need to tune $c$ on a per-game basis, as $c = 0.2$ worked well consistently for all games and the two algorithms demonstrating that obtaining good performance does not hinge on per-game exhaustive search over $c$. We also observed that any value of $c \geq 0.1$ resulted in improvements over vanilla DQN.

## 6.2 RESULTS

The most important question to answer is whether endowing the DQN agent with the proximal term can yield significant improvements over the original DQN. To answer this question in the affirmative, our first result is a large-scale comparison between DQN and DQNPro.

Figure 1 shows a comparison between DQN and DQNPro in terms of the final performance. In particular, for each game, we compute the performance difference between the final policy learned by the DQN agent and the uniformly-random policy, i.e., (DQN − random), as well as (DQNPro − random). Given these two values, we compute the relative improvement of the more performant agent over the weaker one in percentile. Bars shown in red indicate the games in which we observed better final performance for DQNPro relative to DQN, and bars in blue indicate the converse case. The height of a bar denotes the magnitude of this improvement for the corresponding benchmark; notice that the y-axis is scaled logarithmically.

It is clear that DQNPro dramatically improves upon DQN, and this improvement manifests itself in all but 6 games tested. This improvement is particularly rewarding in light of the fact that the implementation of the DQNPro agent is just minimally different than that of the DQN, and that DQNPro adds almost no additional computational burden to the original DQN agent. We also show learning curves on 4 Atari games (Asterix, Breakout, Gravitar, and SpaceInvaders) in Figure 2, which demonstrate that DQNPro typically yields better interim performance as well. We defer to Figure S4 in the Appendix for full learning curves on all games tested.

Can we combine DQNPro with some of the existing algorithmic improvements of DQN? One interesting idea in this space is to bootstrap based on $Q\big(s, \arg\max_a Q(s, a; w); \theta\big)$ as opposed to $\max Q(s, a; \theta)$ during value-function optimization (van Hasselt et al., 2016). Referred to as Double DQN, or DDQN, doing so can hedge against over-estimation issues that arise in Q-learning in light of the convexity of the $\max$ operator (Thrun and Schwartz, 1993). Recall that DQNPro encourages the online network to stay in the vicinity of the target network. As such, it would be interesting to

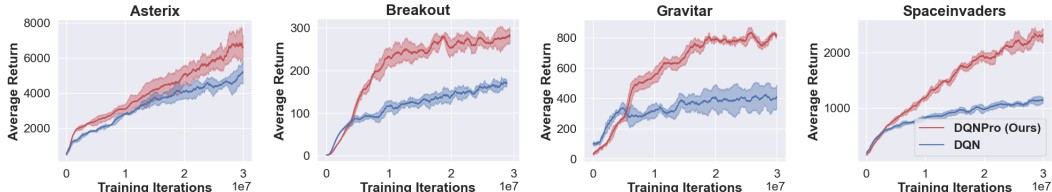

Figure 2: **Learning curves for DQN (blue) and DQNPro (red)** on four Atari games. X-axis indicates the number of steps from the environment used in training and Y-axis shows undiscounted return (sum of rewards). See also Figure S4 for learning curves for all experimental environments.

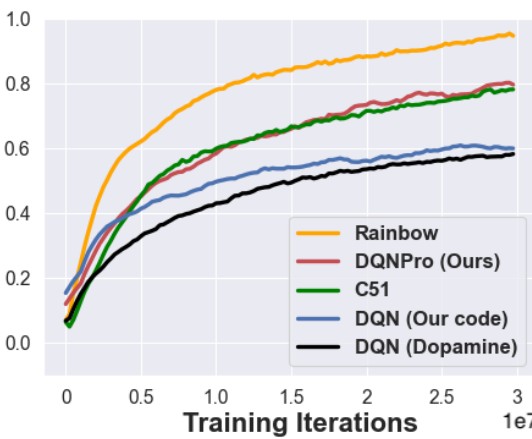

Figure 3: **A comparison between DQNPro and various deep RL baselines**. To compare the agents across all games, we first normalize the performance of all agents so we can average over all games meaningfully. To this end, for each combination of algorithm, game, and number of training iterations, we compute the performance relative to the best policy learned by Rainbow (Hessel et al., 2018). We then average out this quantity across all games.

see if we can still fruitfully combine DDQN and DQNPro since a successful combination hinges on the presence of some discrepancy between the two networks.

In Figure S2, we answer this question affirmatively as well by comparing the final performance of DDQN and DDQNPro on the same set of games tested in the previous experiment, where we used the same values for all hyper-parameters including $c = 0.2$. Notice again, that the value of $c = 0.2$ across all games and for both DQNPro and DDQNPro, demonstrating that this parameter is easy to tune. (See Figure S3 for all learning curves.) Numerous other interesting improvements for DQN exist in the RL literature (Hessel et al., 2018). Our goal was to show such combinations are promising, and not to exhaustively test all of them, so we leave further exploration of other combinations for future work.

Finally, it is valuable to evaluate the performance of DQNPro relative to the more recent baselines. In Figure 3, we provide a comparison between DQNPro, our own DQN implementation, and the Dopamine (Castro et al., 2018) implementation for DQN, C51 (Bellemare et al., 2017), and Rainbow (Hessel et al., 2018). First, observe that our DQN implementation obtains a performance analogous to that of the DQN implementation from Dopamine, thus confirming the soundness of our results. More importantly, our DQNPro algorithm is able to achieve more than 50 percent of the improvement of Rainbow over the original DQN, and is also performing equally well relative to C51. We find this result appealing, because DQNPro is remarkably simpler to understand and to implement relative to baselines such as Rainbow.

### 6.3 ABLATION EXPERIMENTS AND FURTHER ANALYSES

**Effect of the proximal term.** Notice that the purpose of our endowing the agent with the proximal term was to keep the online network in the vicinity of the target network, so it would be natural to ask if this property can be observed in practice. In Figure 4, we answer this question affirmatively by plotting the magnitude of the update to the target network during synchronization. Notice that we periodically synchronize online and target networks, so the proximity of the online and target

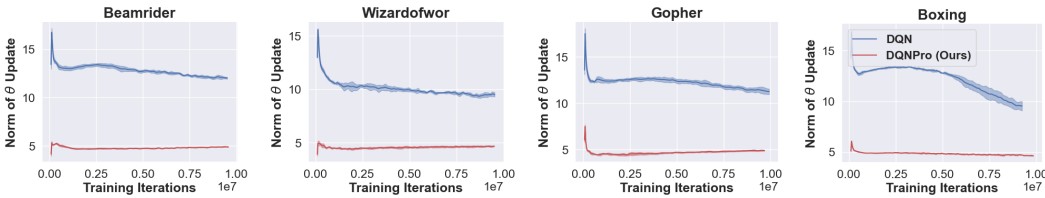

Figure 4: **The effect of the proximal term on the magnitude of updates to the target network.** These results clearly demonstrate that our method has more success than DQN in keeping the online network close to the target network.

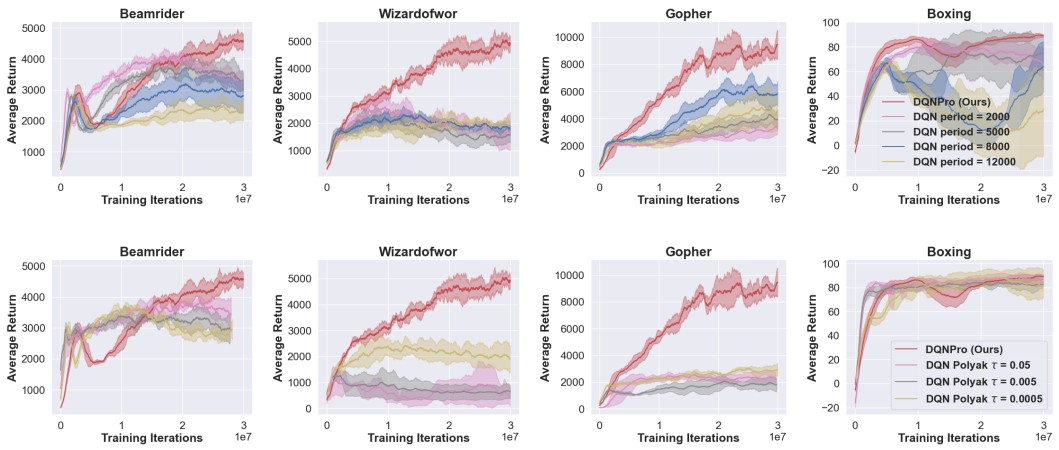

Figure 5: **A comparison between DQNPro and DQN with periodic (Top) and Polyak (Bottom) updates for target network.**

network should manifest itself in a low distance between two consecutive target networks. Indeed, the results demonstrate the success of the proximal term in terms of obtaining the desired proximity of online and target networks.

**Sensitivity to target network update strategy.** While using the proximal term leads to significant improvements when combined with DQN/DDQN, one may still wonder if the advantage of DQNPro over DQN is merely stemming from a poorly-chosen *period* hyper-parameter in the original DQN, as opposed to a truly more stable optimization in DQNPro. To refute this hypothesis, we ran DQN with various settings of the *period* hyper-parameter $\{2000, 4000, 8000, 12000\}$. This set included the default value of the hyper-parameter (8000) from the original paper (Mnih et al., 2015), but also covered a wider set of settings.

Additionally, we tried an alternative update strategy for the target network, referred to as Polyak averaging, which was popularized in the context of continuous-action RL (Lillicrap et al., 2015), and which proceeds by updating the target network as follows: $\theta \leftarrow \tau w + (1 - \tau)\theta$. For this update strategy, too, we tried different settings of the $\tau$ hyper-parameter, namely $\{0.05, 0.005, 0.0005\}$, which includes the value 0.005 used in numerous papers (Lillicrap et al., 2015; Fujimoto et al., 2018; Asadi et al., 2021).

Figure 5 presents a comparison between DQNPro and DQN with periodic and Polyak target updates for various hyper-parameter settings of *period* and $\tau$. It is clear that DQNPro is consistently outperforming the two alternatives regardless of the specific values of *period* and $\tau$, thus clearly demonstrating that the improvement is stemming from a more stable optimization procedure leading to a better interplay between the two networks.

## 7 RELATED WORK

The introduction of the proximal operator could be traced back to the seminal work of Moreau (1962; 1965), Martinet (1970) and Rockafellar (1976), and the use of the proximal operators and related algorithms has since expanded into many areas of science such as signal processing (Combettes and Pesquet, 2009), statistics and machine learning (Beck and Teboulle, 2009; Polson et al., 2015; Reddi et al., 2015), and convex optimization (Parikh and Boyd, 2014; Bertsekas, 2011b;a).

In the context of RL, Mahadevan et al. (2014) introduced a theory of proximal RL for deriving convergent off-policy temporal difference algorithms with linear function approximation. One intriguing characteristic of their family of algorithms is that they perform updates in primal-dual space, a property that was recently leveraged in a finite sample complexity analysis (Liu et al., 2020) for the proximal counterparts of the gradient temporal-difference algorithm (Sutton et al., 2008). Proximal operators also have appeared in recent work in the deep RL literature. For instance, in the context of meta-learning, Fakoor et al. (2020b) used proximal operators during the adaption phase to keep the new parameters close to the meta parameters. Similarly, Maggipinto et al. (2020) improved TD3 (Fujimoto et al., 2018) by employing a stochastic proximal-point interpretation and bootstrapping action value estimates for continuous control.

It is also worthwhile to note that the effect of the proximal term in our work is to ensure that the online network remains in the vicinity of the target network (see also Remark 1), which is reminiscent of the use of trust regions in policy gradient-based methods (Schulman et al., 2015; 2017; Wang et al., 2019; Fakoor et al., 2020a; Tomar et al., 2021). However, three factors differentiate our work: we define the proximal term using the value function, not the policy, we enforce the proximal term in the parameter space, as opposed to the function space, and we use the target network as the previous iterate in our proximal definition.

Finally, existing work in RL showed the benefits of leveraging the structure of the RL problem during optimization. For example, Precup and Sutton (1997) argued that the space of value functions could better be understood as a manifold with an entropic metric, resulting in the exponentiated TD algorithm. Mahadevan and Liu (2012) used mirror descent to penalize weights with large p-norms, a biasing scheme that unfortunately changes the fixed-point of the resultant Bellman equation. Similarly, van Seijen et al. (2019) advocated for performing Q-updates in a logarithmic space to combat issues that arise in problems where the action-gap (Farahmand, 2011) can vary drastically on a state-by-state basis; however, extra work is necessary in their approach to ensure that negative values can be represented.

## 8 CONCLUSION AND FUTURE WORK

We showed a clear advantage in using Proximal Iteration in the context of value-function optimization in deep reinforcement learning. Proximal Iteration ensures that the online network remains in the vicinity of the target network, and shows robustness with respect to noise and stochasticity in reinforcement learning. Several improvements to proximal methods exist, such as the acceleration algorithm (Nesterov, 1983; Li and Lin, 2015), as well as using other proximal terms (Combettes and Pesquet, 2009), which we leave for future work. Our results demonstrate the rewarding nature of developing principled optimization techniques for deep reinforcement learning.

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

# Appendix: Deep Q-Network with Proximal Iteration

## A IMPLEMENTATION DETAILS

Table S1 and Table S2 show hyper-parameters, computing infrastructure, and libraries used for the experiments in this paper for all games tested. Our training and evaluation protocols and the hyper-parameter settings closely follow those of the Dopamine[4] (Castro et al., 2018) baseline. To report performance results, we measured the undiscounted sum of rewards obtained by the learned policy during evaluation.

| DQN and DDQN hyper-parameters (shared) | |
|---|---|
| Replay Buffer size | 200000 |
| Target update period | 8000 |
| Max steps per episode | 27000 |
| Evaluation frequency | 10000 |
| Batch Size | 64 |
| Update period | 4 |
| Number of frame skip | 4 |
| Number of episodes to evaluate | 2 |
| Update horizon | 1 |
| $\epsilon$-greedy (training time) | 0.01 |
| $\epsilon$-greedy (evaluation time) | 0.001 |
| $\epsilon$-greedy decay period | 250000 |
| Burn-in period / Min replay size | 20000 |
| Learning rate | $10^{-4}$ |
| Discount factor ($\gamma$) | 0.99 |
| Total number of iterations | $3 \times 10^7$ |
| Sticky actions | True |
| Optimizer | Adam (Kingma and Ba, 2015) |
| Network Architecture | Nature DQN network (Mnih et al., 2015) |
| Random Seeds | $\{0, 1, 2, 3\}$ |
| Our DQNPro and DDQNPro hyper-parameter | |
| $c$ | 0.2 |

Table S1: **Hyper-parameters used for all methods for all 40 games of Atari-2600 benchmarks**. All results reported in our paper are averages over repeated runs initialized with each of the random seeds listed above and run for the listed number of episodes.

| Computing Infrastructure | |
|---|---|
| Machine Type | AWS EC2 - p2.16xlarge |
| GPU Family | Tesla K80 |
| CPU Family | Intel Xeon 2.30GHz |
| CUDA Version | 11.0 |
| NVIDIA-Driver | 450.80.02 |
| Library Version | |
| Python | 3.8.5 |
| Numpy | 1.20.1 |
| Gym | 0.18.0 |
| Pytorch | 1.8.0 |

Table S2: **Computing infrastructure and software libraries used in all experiments in this paper.**

---

[4]https://github.com/google/dopamine

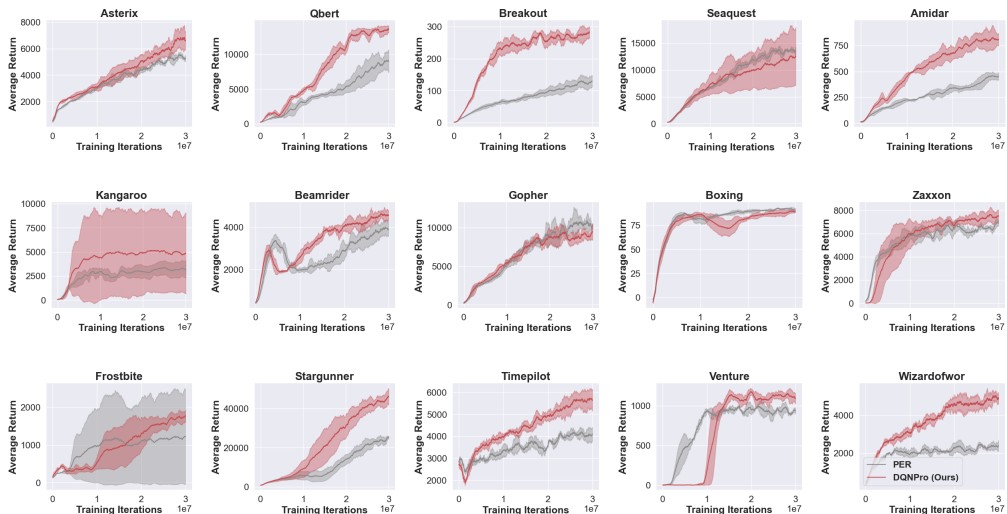

Figure S1: **Learning curves for DQN with prioritized experience replay (PER) (gary) and DQNPro (red)** on 15 Atari games. X-axis indicates the number of steps from the environment used in training and Y-axis shows average undiscounted return.

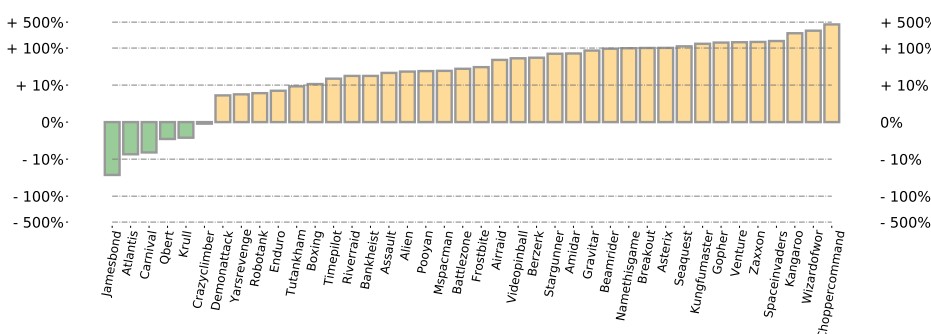

Figure S2: **Performance gain for DDQNPro over the original DDQN in term of the final performance.** Results are averaged over four random seeds. The y-axis is scaled logarithmically.

## B ADDITIONAL EXPERIMENTS

**Prioritized Experience Replay (PER).** In the standard DQN (Mnih et al., 2015), data are uniformly sampled from the reply buffer regardless of their importance. To further improve the performance of DQN, Schaul et al. (2016) proposed an effective way, called prioritized experience replay, to prioritize samples that are more conducive to sample-efficient learning. To further evaluate and situate DQNPro, we compare DQNPro using standard replay buffer against DQN with prioritized experience replay (called PER) on 15 Atari games. As Figure S1 shows our method outperforms PER on the majority of these 15 games. These results provide another data point that our method is a more effective approach for improving the performance of DQN while adding negligible computation cost relative to PER.

## C LEARNING CURVES

In this section, we provide all learning curves for the experiments described in Section 6.2 of the main text. Specifically, Figure S4 shows learning curves for DQN and DQNPro and Figure S2 and Figure S3 show results comparing DDQNPro against DDQN for 40 Atari-2600 games.

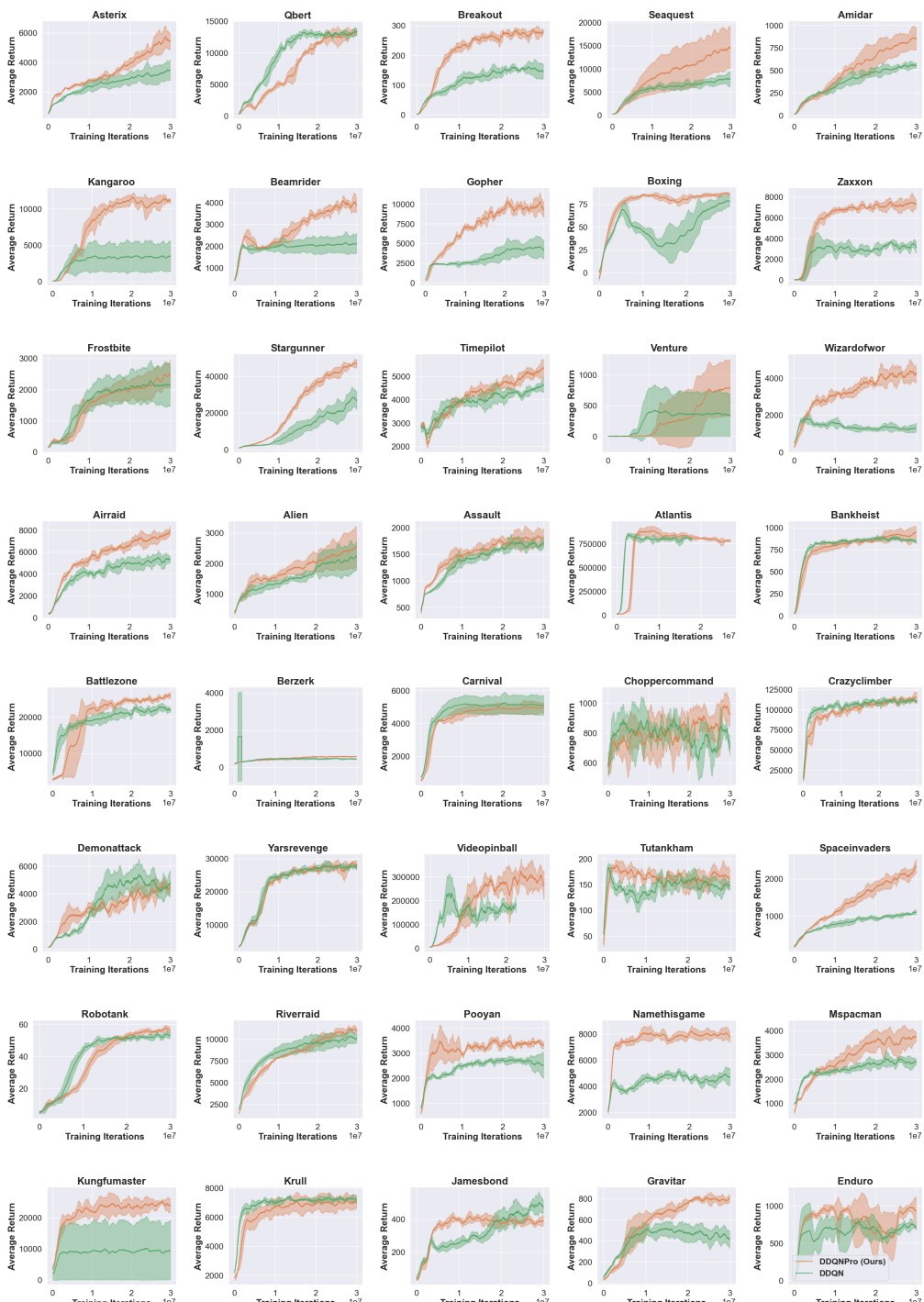

Figure S3: **Learning curves for DDQN (green) and DDQNPro (orange)** on 40 Atari games. X-axis indicates the number of steps from the environment used in training and Y-axis shows average undiscounted return.

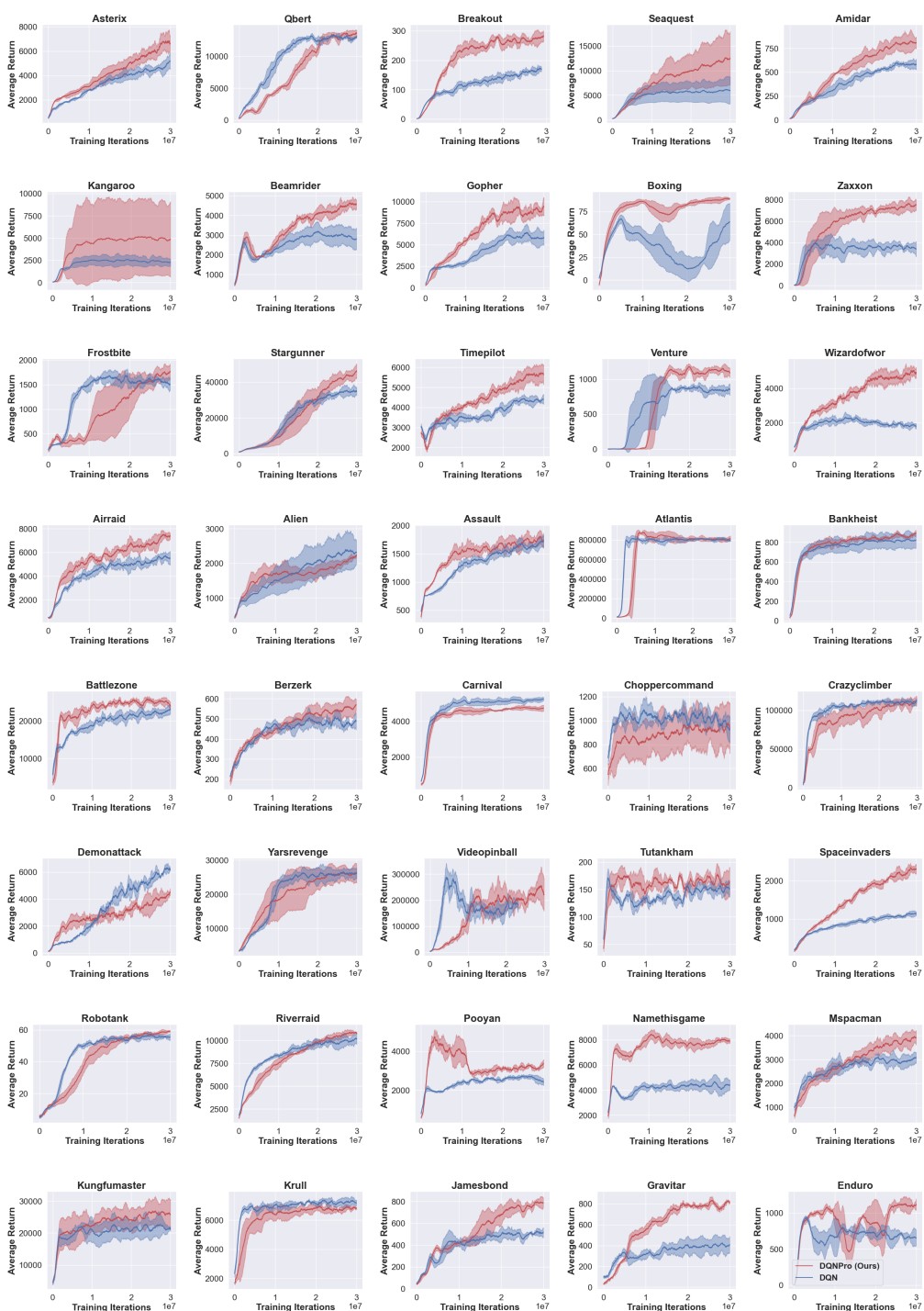

Figure S4: **Learning curves for DQN (blue) and DQNPro (red)** on 40 Atari games. X-axis indicates the number of steps from the environment used in training and Y-axis shows average undiscounted return

Figure S5: A comparison between DQN, DQNPro in parameter space, and DQN in function space.

It would be natural to ask if adding the proximal term in the function space can yield similar improvements. To test this hypothesis, we augment the loss function in DQN by performing the following update given a batch:

$$\nabla h(w) \leftarrow \nabla_w \frac{1}{2|\mathcal{B}|} \sum_{\langle s,a,r,s'\rangle \in \mathcal{B}} \left(r + \gamma \max_{a'} Q(s',a';\theta) - Q(s,a;w)\right)^2 + \frac{1}{2c}\left(Q(s,a;w) - Q(s,a;\theta)\right)^2$$

This will ensure that we still update based on the original loss, but keep the two networks close in the function space. Results shown in Figure S5 compares this idea with the original DQNPro performed in the parameter space. We observe that adding the proximal term in the function space is not nearly as effective as the parameter space. We conjecture that in this case we can only maintain closeness on samples drawn from the batch. In other words, the function are still allowed to differ significantly in all areas of the state-action space but those used for training. This ultimately results in inferior performance. Additional work may be needed to perform proximal gradient steps in the function space, such as through using natural gradients.

ADDITIONAL REFERENCES FOR THE APPENDIX

D. P. Kingma and J. Ba. Adam: A method for stochastic optimization. In International Conference on Learning Representations, 2015.

V. Mnih, K. Kavukcuoglu, D. Silver, A. A. Rusu, J. Veness, M. G. Bellemare, A. Graves, M. Ried-miller, A. K. Fidjeland, G. Ostrovski, et al. Human-level control through deep reinforcement learning. Nature, 2015.

T. Schaul, J. Quan, I. Antonoglou, and D. Silver. Prioritized experience replay. In International Conference on Learning Representations, 2016.

