# OpenReview forum: "Deep Q-Network with Proximal Iteration"
_ICLR.cc/2022/Conference — ICLR 2022 Submitted_

### Official Review · Reviewer_xGvm · 2021-10-30

**Correctness:** 4
**Technical Novelty And Significance:** 3
**Empirical Novelty And Significance:** 3
**Recommendation:** 5
**Confidence:** 5

**Main Review:**

$\textbf{Strengths}$
The method proposed is simple and effective, although proximal iteration is common in RL algorithms, this is the first trial to apply it to the parameters of the critic networks. The authors theoretically analyze the advantage of proximal iteration. The experimental results are solid and the ablation study is carefully conducted. The overall writing and organization are excellent.

$\textbf{Weaknesses}$
The technical contribution is limited given what the authors show in the experiment part, both DQN and DDQN are old models, so I think the authors should try to address at least one of the following aspects.

(1) Apply the proposed technique to more advanced deep reinforcement learning models, like DDPG and SAC on continuous control problems.

(2) Analyze the source of improvement of other DQN variants, in terms of the magnitude of the updates to the target network similar to Figure 4, I think this will give a new perspective into the understanding of the critic network.

**Summary Of The Paper:**

This paper innovatively adds the proximal iteration to the update of the critic network, without the additional computational cost compared with the original SGD optimization method, it empirically demonstrates the advantage of this technique.

**Summary Of The Review:**

This paper is innovative and well written, the experiment parts are very solid and the authors give a clear analysis. But I'm concerned about its technical contribution and conservatively rate it as around the borderline. I hope the authors could address or explain their thoughts about my concerns.

---

> ### Author Response · Authors · 2021-11-18
> **R4**
>
> We appreciate the suggestion for incorporating proximal iteration for continuous control. However, In continuous control, it is non-standard to use periodic target networks. The standard practice is to use Polyak updates for target network, which significantly deviates from the current update regime in Proximal Iteration. While a more universal update may exist that generalizes both regimes, this would be beyond the scope of our paper. If accepted, we would like to focus on this direction for future work.

---

> > ### Comment · Reviewer_xGvm · 2021-11-19
> > **Thanks for the author response**
> >
> > First, thank you for the clarification about the continuous control. I understand this is a simple technique that demonstrates the significant improvement of DQN, but my concern is how you identify the problem of DQN and why it is significant. I think it has not been fully addressed in the response.
> >
> > This paper claims that DQNPro keeps the online network close to the target network, from both the theoretical part and experimental part (like Figure 4), and it shows this technique is beneficial to DQN. My understanding of the problem to address is "SGD will make the online network too far away from the target network and thus hurt the performance". Although DQNPro explicitly applies proximal iteration to solve the problem, could it be possible that this problem has already been mitigated implicitly in the advanced versions of DQN like Rainbow? In other words, if proximal iteration could also improve the performance of Rainbow, then I think the problem addressed by this paper is significant.
> >
> > From the other perspective, even if proximal iteration cannot improve advanced DQNs, you can still show that these advanced DQNs indeed make the online network closer to the target network in each update, and thus bring the improvement, I will still regard your technique as valuable and inspiring for the future work.
> >
> > In the worst case, the advanced versions of DQN have a large gap between the online network and target network in each update, and reducing this gap cannot further improve them, then from my perspective, this technique is not that significant, which can only take effect on some simple models.
> >
> > I hope the authors could think about my questions and let me know if I misunderstood any parts of your work. Basically, we have already had enough techniques, either simple or complex, to improve DQN, so the key point is to propose a technique that is consistently effective for a family of models rather than one specific kind of model.

---

### Official Review · Reviewer_ULKo · 2021-10-31

**Correctness:** 3
**Technical Novelty And Significance:** 2
**Empirical Novelty And Significance:** 2
**Recommendation:** 5
**Confidence:** 4

**Main Review:**

**Strengths:**

This submission is well written and easy to read. The authors conduct extensive experiments to demonstrate the proposed regularization term in proximal iteration boost the performance of different DQN-type algorithms on a lot of Atari games.

**Weakness:**
1. My biggest concern is about the novelty of this submission. Although the regularization term in Equation 4 is effective for DQN-type algorithms, the authors did not give a convincing explanation to explain why proximal iteration works in this setting. In addition, the theoretical analysis in the main text (Remark 1 and 2) are all existing results, and the authors do not introduce new theoretical results to the community.
2. The authors propose to bias the online weights towards the target network, but no detailed explanation is given for this choice. A more natural option is to bias the online weights towards the old weights in the previous iteration, the authors should at least compare the performance of these two options.
3. In Equation 6, $\theta$ is updated every iteration, but in Algorithm 1, $\theta$ is updated only when numUpdates % period = 0.
4. The author just verified through experiments that DQNPro can be combined with DDQN. Since there are some contradictions between the functions of DQNPro and DDQN, is there a deeper explanation for the performance improvement?
5. Minor issue: It is better to redraw some figures in the experiment section using more professional software, Figures 2-5 look blurry.


**Summary Of The Paper:**

This submission exploits *Proximal Iteration* to optimize DQN-type RL algorithms which ensures the online-network remains in the vicinity of the target network to stabilize the training process. This simple *trick* improves the performance of DQN and DDQN on a set of Atari-2600 environments.

**Summary Of The Review:**

The authors use a simple trick to improve the performance of DQN-type algorithms on some Atari environments. Although the proposed method has achieved good results, the novelty is somewhat limited, and the reasons for the performance improvement are not explained clearly.

---

> ### Author Response · Authors · 2021-11-18
> **R3**
>
> The reviewer is concerned about the novelty of our work. While variants of Proximal Iteration have been used in previous work, to the best of our knowledge, we are the first paper to use this technique in DQN. Since Proximal Iteration is very well-studied in optimization, we believed it was sufficient to transparently cite the existing work from the optimization literature.
>
> The reviewer also asks for a comparison between biasing towards the previous target network and biasing towards the previous online network iterate. We remind the reviewer that biasing towards the previous online network iterate is exactly equivalent to SGD. To see this, lets define the following optimization problem:
>
> $$w^{*}=\arg\min w^{T}\nabla f(w_0) + \frac{1}{2\alpha}||w-w_0||^2$$
>
> With simple algebraic calculation, we can deduce that:
> $$ w^{*}= w_{0} - \alpha \nabla f(w_0) $$
> In other words, SGD minimizes the linear approximation of the function plus a proximal term based on the previous online iterate, which could be thought of as the base of the optimizer behind DQN. In other words, using the previous online-network iterate in the proximal term would lead into the original DQN update, which we are already evaluating DQNPro against.

---

> > ### Comment · Reviewer_ULKo · 2021-11-29
> > **I keep my rating unchanged.**
> >
> > Like other reviewers, my biggest concern is the lack of a more in-depth analysis of the effectiveness of DQNPro. While it is nice to improve the performance of DQN by changing a few lines of code, I think it is more important to explain why and when it works. Therefore, I keep my original score.

---

### Official Review · Reviewer_rd5C · 2021-10-31

**Correctness:** 4
**Technical Novelty And Significance:** 3
**Empirical Novelty And Significance:** Not applicable
**Recommendation:** 5
**Confidence:** 4

**Main Review:**

**Strong Points:**

The main strength of this method is in its relative simplicity relative to its empirical performance improvements. Incorporating an L2 weight regularization penalty into existing frameworks is relatively straightforward, as the authors emphasize in Algorithm 1 by modifying a single line. As shown in Figures 1 and S2, this simple change results in a fairly consistent and significant improvement in task performance.

In addition to the conceptual simplicity of the proposed method, its computational cost is also negligible, involving only a squared L2 norm between sets of weights.

While the proposed method introduces an additional hyperparameter, $c$, the authors note that it is relatively easy to tune, setting it to 0.2 for all experiments based on a hyperparameter sweep on four environments. Thus, it seems likely that this method will be fairly straightforward to apply to other environments and algorithmic setups.

The empirical analysis provides performance comparisons with relevant baselines over a substantial portion of the Atari 2600 benchmark, a standard set of environments for pixel-based deep RL with discrete actions. The authors directly compare with both DQN and DDQN, while also providing a comparison with the DQN implementation from the Dopamine codebase. One can conclude from this analysis that the proposed method generally improves performance in these settings. The authors also provide additional analyses, investigating the magnitude of updates, as well as sweeping over various update rates and techniques for DQN.

**Weak Points:**

The primary weak point of this paper is an overall lack of clarity regarding why the proposed method helps. I view this as equally as important, or more, as the empirical evaluation itself. That is, proposing a simple change to deep Q-learning methods may help improve current methods, but understanding why this change helps would improve or unlock a range of future methods, illuminating the root cause of the issue. Currently, the paper alludes to the ‘deadly triad,’ mentioning, but not explaining, “extreme volatility in the optimization procedure.” The authors go on to propose proximal iteration as a method for combatting such volatility, however, as the authors acknowledge, these intuitions are not necessarily applicable in the case of non-linear function approximators. Indeed, the deadly triad manifests in issues with estimating Q-values, e.g., optimism bias. Yet, with non-linear function approximators, proximal iteration in the space of weights provides no guarantees on the stability of Q-value outputs. To properly situate their method, the authors would need to explain a) what is unique about the “extreme volatility” that arises in the deadly triad, b) why it is appropriate to apply proximal iteration in the weight-space rather than the output space, e.g., Piche et al., 2021.


The proposed method is somewhat lacking in novelty, and the authors are missing a substantial number of citations of previous works that have investigated constraining parameters or outputs in value networks. Examples include natural gradient deep Q-learning (Knight & Lerner, 2018), PreQN (Achiam et al., 2019), KOVA (Di-Castro Shashua & Mannor, 2020), GAE (Schulman et al., 2015), etc. While the proposed method does not exactly match the setups in these previous works, the motivation and overall idea are highly similar. At the very least, a more in-depth discussion of related work is required. Further, given the similarity with these previous works, it would be helpful to compare with some other stabilization schemes.

The authors investigate their method entirely in the Atari domain. Accordingly, it’s unclear whether the hypothesized instabilities in Q-learning are more pronounced in these environments and whether the proposed method is generally applicable outside of this domain. While I do not see this as grounds for rejection, experiments in other domains would help to bolster the authors’ claims.

Likewise, because the authors only evaluate on Atari, their method is entirely evaluated on discrete control tasks, whereas Q-networks are also utilized in continuous control tasks. To truly demonstrate the generality of the problem and proposed solution, it would be helpful if the paper included at least some experiments with continuous control experiments, e.g, with actor-critic algorithms. The proposed method could also be applied for the purposes of state-value estimation.


- Piche et al., 2021. Beyond target networks: improving deep q-learning with functional regularization
- Knight & Lerner, 2018. Natural gradient deep q-learning
- Achiam et al., 2019. Towards characterizing divergence in deep q-learning
- Di-Castro Shashua & Mannor, 2020. Kalman meets Bellman: improving policy evaluation through value tracking
- Schulman et al., 2015. High-dimensional continuous control using generalized advantage estimation.


**Additional Comments:**

Some of the background in sections 2 and 3 feels overly pedagogical and/or not applicable to the setting used in the paper. For instance, it’s not clear what the discussion of the contraction mapping really adds to the paper. Similarly, much of the paper is devoted to discussing proximal methods in the context of strongly convex objectives, e.g., with linear functions, which is not the setting used in the paper. This space would be better utilized by characterizing the instabilities with Q-learning and explaining why proximal iteration in weight space is appropriate for tackling these instabilities.

Equation 6: I believe the last line shouldn’t be indented. Also, this feels somewhat redundant, given that it’s a simplified version of Algorithm 1.

Results: I would present the DDQN results in the main paper, perhaps right alongside the DQN results. It’s not clear how the 40 environments were selected from the total set of environments. The learning curves in Figure 2 are cherry-picked. It’s unclear what the update magnitudes tell us about the algorithms. These results need to be tied more specifically to the issues with deep Q-learning.


**Summary Of The Paper:**

This paper applies proximal iteration, via L2 regularization from the weights of the target network, to deep Q-learning. Although the change in the algorithm is minimal, the authors demonstrate performance improvements across Atari environments as compared with baselines.

**Summary Of The Review:**

My reasons are outlined in the weak points section above. While this paper provides a set of promising empirical results, the relative lack of explanation and analysis around why proximal iteration helps in deep Q-learning significantly detracts from the impact of this paper. I would like to see these aspects of the paper improved, perhaps drawing on previous works that have tackled this problem. Relatedly, given that other works have attempted to improve deep Q-learning via regularization, these works need to be cited and, ideally, compared against. Finally, it would help to expand the scope of the results section to include other environments or setups in which deep Q-learning is used, e.g., actor-critic algorithms, demonstrating the generality of the proposed method. In its current form, this paper will leave readers a) confused regarding what problems the proposed method is tackling, b) unaware of previous works in this area, and c) unsure about the method’s generality. For these reasons, I cannot recommend acceptance.

---

> ### Author Response · Authors · 2021-11-18
> **R2**
>
> The reviewer is concerned with insufficient justification for the benefits of Proximal Iteration. In this paper we leaned on the existing results from the optimization literature that already clearly demonstrated the benefits of Proximal Iteration over SGD [1]. further study of Proximal Iteration seemed redundant in this case in light of the existence of a rich body of literature on Proximal Iteration.
>
> That said, we agree with the reviewer that we could have better situated our work within the RL literature, which would have better emphasized the advantages of Proximal Iteration in the context of RL with function approximation. We are happy to add citations to the papers the reviewers provide and will add a discussion. It is also worthwhile to note that we have done extensive experiments for performing a similar regularization in the function space both before and after the submission. In our experience, doing the regularization in the function space is not effective at all relative to the parameter space. Please see our new experiment on page 18 in the Appendix.
>
> [1] Ernest K. Ryu, Stephen Boyd, “Stochastic Proximal Iteration: A Non-Asymptotic Improvement Upon Stochastic Gradient Descent”

---

> > ### Comment · Reviewer_rd5C · 2021-11-27
> > **Response to authors**
> >
> > Thank you for your response.
> >
> > To clarify, my concern is not with insufficient justification for the benefits of proximal iteration, as the authors state. I am fully convinced that proximal iteration is a useful technique in general. Instead, I am concerned with the paper's lack of justification for using (weight-space) proximal iteration in deep Q-learning algorithms. This isn't an issue of providing theoretical proofs or guarantees. Rather, this is about properly understanding the issue that's being addressed by this method. This is an important consideration for any paper, both theoretical or empirical. The paper is unclear or relatively lacking in addressing the following questions:
> > - *Why is proximal iteration specifically helpful in the case of TD-learning?* In other words, what is it about TD-learning that requires proximal iteration compared with, e.g., supervised learning? The paper currently alludes to the 'deadly triad,' but the reader is left to guess why this is an issue. Toy experiments in more tractable domains may help to illuminate the issue.
> > - *Why is using proximal iteration in weight space an appropriate approach for addressing this issue?* Many proximal methods within RL operate in function-space, e.g., via KL penalties on the policy. What is unique about this situation that necessitates weight-space proximal iteration?
> >
> > This criticism isn't about improved theoretical claims; it's about a more complete understanding of the problem and solution space.
> >
> > Finally, I would like to reiterate some of the other reviewers and state that it would be somewhat helpful to analyze proximal updates in the context of more recent baseline methods. Given that the proposed method is trivial to implement, it should be straightforward to run these comparisons. This will also help to illuminate whether the issues being addressed by DQNPro are orthogonal to those addressed by previous works.

---

### Official Review · Reviewer_UNLa · 2021-11-01

**Correctness:** 3
**Technical Novelty And Significance:** 2
**Empirical Novelty And Significance:** Not applicable
**Recommendation:** 3
**Confidence:** 4

**Main Review:**

The authors point out the various challenges in stabilizing deep reinforcement learning. They argue that applying the idea of proximal iteration to DQN can improve the stability of the learning and the performance of the algorithm. In practice, this implies a small change in the weight update term of the DQN algorithm, which biases the weight changes towards the target network.

The authors compare the resulting algorithm with the original DQN both the version from Google Dopamine and the authors' own reimplementation of it. The approach improves on the original DQN, but it does not matching the current state of the art.

As a note: Formula 6 and algorithm 1 are different - formula 6 does not capture the fact that the \theta is updated only very rarely.

**Summary Of The Paper:**

The paper proposes a modification of the DQN algorithm by adding a proximal term that biases the learning to retain the Q algorithm in the proximity of the target network. The authors show that the proposed algorithm perform better compared to the original DQN algorithm on several Atari games.

**Summary Of The Review:**

The paper describes a relatively simple modification of the DQN algorithm using a proximal term. The approach improves the performance of DQN, but it does not improve on the current state of the art.

---

> ### Author Response · Authors · 2021-11-18
> **R1**
>
> The reviewer seems to advocate for rejection because DQNPro is not improving over Rainbow. To be clear, the Rainbow algorithm is combining various existing algorithmic ideas to improve the original DQN. Rainbow is, therefore, complicated to implement and to analyze. In this work, our goal was not to improve over Rainbow, but to show that DQNPro can significantly improve DQN with a simple proximal modification, and that this improvement roughly covers 60 percent of the gap between DQN and Rainbow. We plotted the Rainbow result for completeness and full transparency. Note that this has been a standard practice in recent works on DQN, see for example van Seijen et. al [1] Figure 9.
>
> [1] Harm van Seijen, Mehdi Fatemi, Arash Tavakoli, “Using a Logarithmic Mapping to Enable Lower Discount Factors in Reinforcement Learning”

---

> > ### Comment · Reviewer_UNLa · 2021-11-29
> > **Retaining the score**
> >
> > Well, at a top venue, one needs to show that the proposed approach is improving on the state of the art along at least some dimensions. This can be absolute performance, faster convergence, lower energy consumption etc. It is possible that the proposed approach does improve on some metric, which I would recommend the authors to investigate for future resubmissions. The fact that it requires a simple change from the DQN source code is not a sufficient reason to be recommended for publication.
> >
> > Thus, I retain my ranking.

---

### Author Response · Authors · 2021-11-18
**General Comment**

We thank all reviewers for their constructive feedback.

Overall, we believe that reviewers have understood the significance of our empirical contribution. To reiterate, our results clearly demonstrate the benefits of using proximal iteration for DQN. In particular, our novel algorithm, DQNPro, significantly outperforms DQN, and even goes as far as outperforming more modern baselines such as C51. This is true despite the fact that our DQNPro is minimally modifying the original DQN, and thus is simple to understand and implement. We strived for simplicity in our contribution and presentation, as we believed simplicity to be a good feature of our work not a bug.

Reviewers, however, raised some concerns pertaining to the theoretical contribution of our work. While we agree that the paper can be strengthened theoretically, we would like to respectfully remind the reviewers that significant empirical contribution is enough to be given a high priority for acceptance as communicated by ICLR program chairs:

“This year, we have made changes to the review forms: reviewers are asked to assess correctness, technical novelty and significance (algorithms, models, and theories),  as well as empirical novelty and significance (advancements, insights, or datasets) separately. Submissions with significant contributions in either technical aspects or empirical aspects will be given high priority for acceptance.”

---

### Decision · Program_Chairs · 2022-01-20

**Decision:**

Reject

**Comment:**

The paper applies proximal iteration to Q-learning, which significantly improves the performance of DQN. Reviewers agreed the paper is not ready for publication, for a couple reasons. DQN is quite far from current state-of-the-art. Improvements therefore need to be well-founded to be of broad interest. If the algorithm that is being improved is not competitive, there should be more general lessons that can be extracted from how and why the improvement works. Unfortunately, the reviewers felt that there was insufficient understanding of why proximal iteration helps.